# Feasibility and Implementation of Ex Vivo Fluorescence Confocal Microscopy for Diagnosis of Oral Leukoplakia: Preliminary Study

**DOI:** 10.3390/diagnostics11060951

**Published:** 2021-05-26

**Authors:** Veronika Shavlokhova, Christa Flechtenmacher, Sameena Sandhu, Michael Vollmer, Andreas Vollmer, Maximilian Pilz, Jürgen Hoffmann, Oliver Ristow, Michael Engel, Christian Freudlsperger

**Affiliations:** 1Department of Oral and Maxillofacial Surgery, University Hospital Heidelberg, 69120 Heidelberg, Germany; Sameena.Sandhu@med.uni-heidelberg.de (S.S.); Michael.Vollmer@med.uni-heidelberg.de (M.V.); andreas.vollmer@med.uni-heidelberg.de (A.V.); Juergen.Hoffmann@med.uni-heidelberg.de (J.H.); Oliver.Ristow@med.uni-heidelberg.de (O.R.); Michael.Engel@med.uni-heidelberg.de (M.E.); Christian.Freudlsperger@med.uni-heidelberg.de (C.F.); 2Department of Pathology, University Hospital Heidelberg, 69120 Heidelberg, Germany; Christa.Flechtenmacher@med.uni-heidelberg.de; 3Department of Medical Biometry at the Institute of Medical Biometry and Informatics, University Hospital Heidelberg, 69120 Heidelberg, Germany; pilz@imbi.uni-heidelberg.de

**Keywords:** leukoplakia, ex vivo fluorescence confocal microscopy, confocal, oral cancer

## Abstract

Background: Oral leukoplakia is a potentially malignant lesion with a clinical impression similar to different benign and malignant lesions. Ex vivo fluorescence confocal microscopy is a developing approach for a rapid “chairside” detection of oral lesions with a cellular-level resolution. A possible application of interest is a quick differentiation of benign oral pathology from normal or cancerous tissue. The aim of this study was to analyze the sensitivity and specificity of ex vivo fluorescence confocal microscopy (FCM) for detecting oral leukoplakia and to compare confocal images with gold-standard histopathology. Methods: Imaging of 106 submosaics of 27 oral lesions was performed using an ex vivo fluorescence confocal microscope immediately after excision. Every confocal image was qualitatively assessed for presence or absence of leukoplakia by an expert reader of confocal images. The results were compared to conventional histopathology with H&E staining. Results: Leukoplakia was detected with an overall sensitivity of 96.3%, specificity of 92.3%, positive predictive value of 93%, and negative predictive value of 96%. Conclusion: The results demonstrate the potential of ex vivo confocal microscopy in fresh tissue for rapid real-time assessment of oral pathologies.

## 1. Introduction

Oral leukoplakia (OL) is the most frequent potentially malignant disorder of the oral mucosa [1,2]. The term leukoplakia is defined by the World Health Organization (WHO) as “a white plaque of questionable risk having excluded (other) known diseases or disorders that carry no increased risk for cancer” [3]. The rate of malignant transformation of OL varies between 0.13 and 36.4% [4] in the literature, depending on study design and investigated population, with a reported annual rate of up to 2.9% [5,6].

Although several clinical, histological, and molecular features have been shown to correlate with a risk for malignant transformation, the presence and degree of epithelial dysplasia based on histopathological evaluation is still considered the most common objective in the management of oral leukoplakia. According to the WHO classification of OL based on architectural and cytological features, the grading of epithelial dysplasia is divided into three categories: mild, moderate, and severe. Leukoplakia with no or simple dysplasia (low malignant risk) may be either removed or not, depending on location, size, and presence of risk factors (e.g., smoking anamnesis) [7]. In the presence of moderate or severe epithelial dysplasia, surgical treatment is recommended [8]. In more than 10% of cases, the reoccurrence of the lesion after removal has been reported [9].

According to the classification of OL by van der Waal [10], the stage of leukoplakia depends on the size, clinical appearance (homogeneous/non-homogeneous), and histology (occurrence of dysplasia). Non-homogeneous lesions with dysplasia are more likely to undergo malignant transformation.

Although biopsy and histopathological investigation are accepted as the gold standard method of diagnosis of OL, new technologies with a cost- and time-saving potential to determine the possible presence and degree of epithelial dysplasia “chairside” are of growing importance for future clinical implications. Furthermore, the ongoing development and increasing importance of computer-aided surgery and application of artificial intelligence in medicine prove the need for new technologies. In particular, the ability to obtain high-resolution digital sample images is one important step in regards to computer-aided or artificial intelligence-based diagnosis.

One such new emerging imaging technology is confocal microscopy. This optical technique enables a rapid in vivo and ex vivo examination of skin and mucosal lesions [11,12,13,14,15,16]. The ex vivo fluorescence confocal microscopy images of skin tissue have been shown to correlate excellently with conventional histopathology and provide images of optical sectioning of specimens with high resolution and without tissue damage [14]. The technology of ex vivo FCM combines two different lasers with wavelengths of 488 nm (fluorescence mode) and 785 nm (infrared, for reflectance mode) [17]. A fluorescent contrast agent (acridine orange) stains nuclei and increases the contrast of nucleated cells which are highlighted by the blue laser. Additionally, the reflectance laser is used for structural information of the sample. The built-in algorithm translates the signal into pseudocolored images very similar to H&E staining [18]. Confocal mosaics display up to 25 × 25 mm of tissue in less than 1 min that corresponds to a view with 2.5× magnification in standard light microscopy. Although the technology enables a real-time assessment of freshly excised tissue, it also needs validation as an investigation tool that is entirely different from classic histopathology.

Staining with acridine orange increases the contrast of the cell nuclei. This fluorescent contrast agent is cell-permeant and binds nucleic acid and dye that emits green fluorescence in the blue laser. Differences in the reflection and backscattering of light are used to obtain and record information on the cellular structure via the reflectance laser. The signal imaging is translated into digitally red–blue colored images, similar to histological H&E staining [18]. An area up to 25 × 25 mm can be scanned in less than 1 min. The resulting confocal mosaic image of the investigated sample corresponds to a 2.5× magnification with light microscopy. The technology is principally different from classic histopathology and still needs to be validated for that purpose, but it has the potential of quick real-time sample investigation.

Recent studies have demonstrated an excellent correlation of skin tissue ex vivo FCM with conventional histopathology [11,12,13,14]. Additionally, the potential of intraoperative use of ex vivo FCM to assist during Mohs surgery in rapid detection of residual tumor tissue in the marginal region was reported in recent studies [19]. Until now, to the best of the authors’ knowledge, no studies show an ex vivo FCM application in oral leukoplakia.

The primary aim of this study was to validate the diagnostic accuracy of ex vivo FCM compared to gold-standard histopathology in terms of oral leukoplakia and to calculate prospectively the sensitivity (S), specificity (Sp), positive predictive value (PPV), and negative predictive value (NPV) in detecting leukoplakia. A secondary aim was to describe ex vivo confocal cytoarchitectural findings in oral mucosae affected by leukoplakia.

## 2. Materials and Methods

From September 2019 to January 2020, a single-institutional observational cohort study with a prospective design was performed. This study was reviewed and accepted by the ethics committee for clinical studies of the Heidelberg University (registry number S-665-2019). The inclusion criteria were (1) patients aged 18 years and older who gave a written informed consent (or their parents if a patient was younger than 18 years), (2) patients who were diagnosed as having a suspicious oral mucosal lesion, and (3) patients for whom a biopsy or resection of the lesion was indicated. The exclusion criteria were (1) recurrent pathology of oral mucosa and (2) previous surgery or other treatments. Twenty-two people with single or multiple suspicious white areas in the mouth with indication for sampling were selected, and a total of 27 oral lesions were identified, described clinically (homogeneous/non-homogeneous), and surgically removed. Immediately after resection, the mucosal excisions were rinsed in isotonic saline solution, immersed in 1 mM acridine orange solution for 20 s, and rinsed again for another 5 s. The examination was then done using an ex vivo FCM (Vivascope 2500 Multilaser, Lucid Inc., Rochester, New York, NY, USA) in combined reflectance and fluorescence modes. Details of the applied confocal mosaic processing and acquisition have been described earlier [14,20]. Acridine orange indicates the nuclei and provides a strong nuclear-to-cytoplasm and nuclear-to-dermis contrast, and the use of it does not affect histopathology, as shown in a study by Gareau et al. [14]. Immediately after completing this step, the samples were transferred to our pathological department for conventional histopathological examination. The mosaics were displayed on a large monitor with high resolution to mimic the standard 2.5× view of histopathology. Twenty-seven mosaics with both positive and negative cases of oral leukoplakia were prepared for this study. Mosaics were divided into four smaller submosaics to provide morphologic characteristics at higher resolution and magnification. All images were reviewed blindly to histopathological diagnoses and clinical appearance data. Submosaics were zoomed in for a precise examination of images at a magnification of 30 to 40×.

Each mosaic was evaluated for presence or absence of dysplasia based on its known microscopic features. Confocal features of dysplasia in leukoplakia have been previously described in an in vivo study by Maitland et al. [21].

All analyses in the current study were performed using SPSS 25.0 (IBM, Armonk, New York, NY, USA) and R version 3.6.2 [22]. Results are sensitivity, specificity, positive predictive value, and negative predictive value for the evaluation of the presence or absence of leukoplakia in single ex vivo confocal submosaic images.

True positive was defined as the presence of morphologic criteria for leukoplakia in both the histopathology and confocal images, true negative was defined as no such findings in both, false positive was defined as the presence of morphologic criteria for leukoplakia in confocal imaging but no such findings in the histopathology, and false negative was defined as the presence of these findings in the histopathological sections but with none in confocal imaging. Microscopic features were compared descriptively using a chi-squared test.

## 3. Results

### 3.1. Detection of Oral Leukoplakia in Ex Vivo Confocal Images

In total, 106 ex vivo images of 27 oral lesions from the 22 patients enrolled in the study were obtained. For each FCM lesion, corresponding H&E histopathology was completed. Each confocal mosaic image was divided into four submosaics and displayed with an approximate magnification of 550×. An example is presented in Figure 1a,b. From 27 mosaics, 106 submosaics were created: 54 (51%) contained leukoplakia and 52 (49%) did not.

Table 1 summarizes patient demographics and their associated lesion characteristics.

Table 2 presents the results of the statistical analysis and evaluation of confocal mosaics compared to the corresponding H&E histopathology. The overall S, Sp, PPV, and NPV in detecting leukoplakia in surgical samples were 96.3%, 92.3%, 93%, and 96%, respectively.

### 3.2. Ex Vivo Confocal Features of Oral Leukoplakia

Histologically, oral leukoplakias vary from hyperkeratosis to mild, moderate, and severe dysplasia to carcinoma in situ and include changes in architectural and cytological features. Architectural features of dysplasia are asymmetrical epithelial stratification, an increased number of mitotic figures in the epithelium, dyskeratosis, drop-shaped rete pegs, keratin pearls within these rete pegs, loss of polarity of basal cells, and basal cell hyperplasia or anaplasia [23]. The cytological features of dysplasia are nuclear pleomorphism, cellular pleomorphism, increase in nuclear–cytoplasmic ratio, prominent nucleoli, and hyperchromasia [23]. In contrast to malignant oral lesions, these changes do not exceed the basal membrane.

Table 3 shows presence or absence of those histological (architectural and cytological) features of leukoplakia compared to normal mucosa in ex vivo fluorescence confocal images.

## 4. Discussion

In our study, high diagnostic accuracy between ex vivo FCM and histopathology was demonstrated. This finding is consistent with previous studies evaluating the accuracy of ex vivo fluorescence confocal microscopy in other histopathological entities and on other anatomical sites [9,11,12,13,14].

Previous studies investigated the use of in vivo RCM for imaging of skin lesions and generated interest in applying this technology to lesions in the head and neck region. In the present study, we were able to demonstrate that the accuracy in diagnosing oral leukoplakia based on ex vivo FCM correlated highly with diagnosis based on gold-standard histopathology.

This finding shows ex vivo FCM to be a clinically applicable alternative method for digital diagnosis of mucosal dysplasia.

Another important finding of the present work was that the known histopathological cytoarchitectural features of dysplasia were also present in all corresponding confocal mosaics. It is known that the degree of epithelial dysplasia is an important risk factor for malignant transformation of OL.

The optimal treatment of high-grade dysplasia is surgical removal. This recommendation is based on the possibility of low to intermediate spontaneous regression rates ranging from 9 to 45% [24].

Management of moderate dysplasia remains controversial as it shows an intermediate propensity to progress to malignancy. Although continuous exposure to known risk factors such as smoking and alcohol can accelerate the progress, there are no biomarkers that can generally predict malignant transformation. Furthermore, no prospective randomized controlled trials have been conducted to determine the optimal management.

For low-grade dysplasia, either observation or resection might be applied. However, in a study by Arnaoutakis et al. [25], the authors found that no clinical differentiation between high-grade, moderate, and low-grade dysplasia is possible. A passive observation (wait and watch) even in case of mild dysplasia is therefore not acceptable.

Hence, histopathological examination still remains the only useful option to determine further treatment. Ex vivo FCM proved to be a useful up-to-date alternative for immediate diagnosis and evaluation of suspect lesions and tissue margins and the determination of optimal treatment. The possibility of chairside application avoids long waiting periods for histopathological results, which can take up to 7 days, depending on the location of the nearest pathological laboratory. Next to being timesaving, a pleasant side-effect is also the avoidance of psychological burden for the patients during an insecure waiting period, especially in cases that undergo multiple samplings over an extended period of time. One major advantage is the possibility of obtaining high-resolution images that can provide a basis for ongoing and future projects in computer-aided diagnosis via artificial intelligence. This can be a valuable future chairside tool, especially for surgeons working in remote areas and having limited access to pathological laboratories.

Another point to mention is the cost-saving potential of FCM compared to that of conventional histopathology. A possible issue to compare is the cost of a single biopsy. For histopathological examination, the costs highly depend on the investigated sample, its size, and its location and range from EUR 50 to 500 (University Hospital Heidelberg). There are still no established official billing items for an ex vivo FCM investigation, but we assume that the costs of a single biopsy could be significantly less or relatively low considering the absence of a tissue preprocessing step and the quickness of the procedure. Additionally, the imaging and image interpretation can be performed by an experienced clinician with no need for an extra pathologist.

One of the limitations of our study was the sample size of 27 oral lesions from 22 patients. This was partly solved by dividing each confocal mosaic into four submosaics to enlarge the number of images. Part of the mosaics could not be assessed due to false positioning for scanning, so another limitation is the technique of sample scanning. Although ex vivo fluorescence confocal microscopy allows an evaluation of the entire lesion in maximal dimensions of 28 × 28 mm, positioning of a sample is still a major challenge for a correct diagnosis. In our study, no standardized orientation of tissue samples during scanning was performed. Expectedly, samples placed and scanned tangentially in ex vivo FCM could not be diagnosed correctly. All samples scanned over the vertical cut through all skin layers were diagnosed correctly.

For the statistics, multiple samples of submosaics were taken from the same patient. These samples were assumed to be independent, which is valid under the assumption that intraindividual variations of the morphologic criteria for leukoplakia are more relevant for the classification performance than interindividual variations. Studies with greater cohorts will be needed in future attempts to reproduce the results with single samples per patient in order to validate this assumption.

The strength of the present study is the inclusion of samples with normal mucosa and corresponding histopathology to each confocal image.

As a secondary finding, our results further demonstrate that the known histological features (architectural and cytological) can also be identified in ex vivo confocal images if samples are assessed properly.

## 5. Conclusions

Ex vivo FCM provides digital high-resolution imaging of leukoplakia that is comparable to H&E but, expectedly, less effective at the moment. Our results demonstrate that the classic histological patterns of dysplasia can also be found in ex vivo pictures and be supportive in the diagnosis of oral leukoplakia. Prospectively, this technology might be supportive in the differentiation of moderate and severe leukoplakia, given a correct positioning of samples in the scanner of ex vivo FCM. Another point to mention is the cost-saving potential of the technology.

## Figures and Tables

**Figure 1 diagnostics-11-00951-f001:**
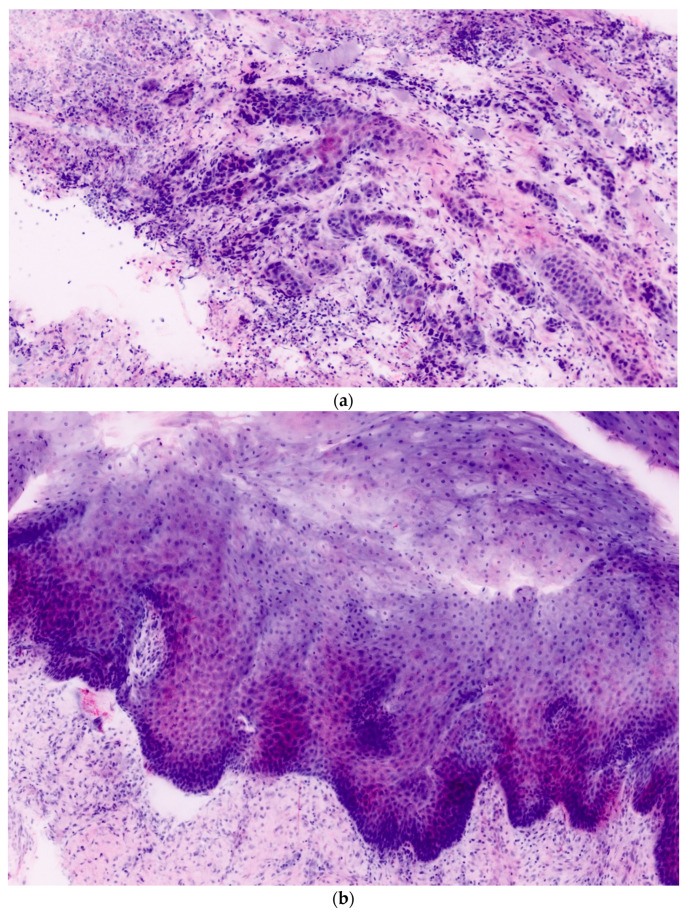
(**a**,**b**) Examples of ex vivo confocal microscopy mosaics of leukoplakia.

**Table 1 diagnostics-11-00951-t001:** Patient (N = 22), lesion (N = 27), and confocal submosaic (N = 106) characteristics.

Patients		
Age	Average (y)	63.7
Gender, N = 22	Female	12 (56%)
	Male	10 (44%)
Submosaic, N = 106	Leukoplakia	54 (51%)
	Normal mucosa	52 (49%)
Dysplasia grade of oral leukoplakia in confocal submosaics, N = 54	Mild and moderate	38 (70%)
	Severe	16 (30%)
Location, N = 27	Lip	1
	Palate	1
	Tongue	7
	Buccal mucosae	17
	Floor of mouth	1

**Table 2 diagnostics-11-00951-t002:** Detection of leukoplakia in confocal submosaics compared to normal mucosa.

	H&E +	H&E −	N (%)
Ex vivo FCM +	52	4	56 (53%)
Ex vivo FCM −	2	48	50 (47%)
N (%)	54 (51%)	52 (49%)	106 (100%)

**Table 3 diagnostics-11-00951-t003:** Microscopic features of dysplasia in ex vivo FCM pictures of leukoplakia (N = 54) and normal mucosa (N = 52).

Microscopic Features of Dysplasia	Presence in Ex Vivo Confocal Submosaics of Leukoplakia (%)			Presence in Ex Vivo Confocal Submosaics of Normal Mucosa (%)			*p*-Value (Presence in Ex Vivo Confocal Submosaics of Leukoplakia vs. Normal Mucosa)
	Present	Not Present	Cannot Be Assessed	Present	Not Present	Cannot Be Assessed	
Asymmetrical epithelial stratification	25 (46.3%)	4 (7.4%)	25 (46.3%)	1 (1.9%)	12 (23.1%)	39 (75%)	4.5886914 × 10^−7^
Increased number of mitotic figures in the epithelium	21 (38.9%)	18 (33.3%)	15 (27.8%)	0 (0%)	30 (57.7%)	22 (42.3%)	3.2146347 × 10^−6^
Dyskeratosis	22 (40.7%)	5 (9.3%)	27 (50%)	0 (0%)	1 (1.9%)	51 (98.1%)	1.1114018 × 10^−7^
Drop-shaped rete pegs	7 (13%)	18 (33.3%)	29 (53.7%)	0 (0%)	13 (25%)	39 (75%)	0.00984
Keratin pearls within these rete pegs	8 (14.8%)	27 (50%)	19 (35.2%)	0 (0%)	18 (34.6%)	34 (65.4%)	9.0618434 × 10^−4^
Loss of polarity of basal cells	25 (46.3%)	6 (11.1%)	23 (42.6%)	1 (1.9%)	8 (15.4%)	43 (82.7%)	6.5653921 × 10^−7^
Basal cell hyperplasia or anaplasia	25 (46.3%)	5 (9.3%)	24 (44.4%)	0 (0%)	10 (19.2%)	42 (80.8%)	1.4098299 × 10^−7^
Nuclear pleomorphism	29 (53.7%)	12 (22.2%)	13 (24.1%)	1 (1.9%)	32 (61.5%)	19 (36.5%)	1.2949366 × 10^−8^
Increase in nuclear–cytoplasmic ratio	38 (70.4%)	7 (13%)	9 (16.7%)	0 (0%)	36 (69.2%)	16 (30.8%)	1.20059 × 10^−13^
Prominent nucleoli	41 (75.9%)	6 (11.1%)	7 (13%)	1 (1.9%)	35 (67.3%)	16 (30.8%)	3.2524771 × 10^−14^
Hyperchromasia	44 (81.5%)	3 (5.6%)	7 (13%)	6 (11.5%)	31 (59.6%)	15 (28.8%)	1.2413161 × 10^−12^
Inflammatory cells	37 (68.5%)	11 (20.4%)	6 (11.1%)	10 (19.2%)	34 (65.4%)	8 (15.4%)	1.0549835 × 10^−6^

## Data Availability

The data presented in this study are available on request from the corresponding author.

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
