# Peer review of "Feasibility and Implementation of Ex Vivo Fluorescence Confocal Microscopy for Diagnosis of Oral Leukoplakia: Preliminary Study"

_diagnostics, 2021, doi:10.3390/diagnostics11060951_

Round 1

Reviewer 1 Report

Dear authors,

the paper - which I have already reviewed in your previous submission - has been significantly improved and is of a certain interest for a journal such as diagnostic. Study design and methods are adequate. It should be accepted in my opinion but I would improve/enlarge the discussion part in which author hypothesize future application for this technique (which, as they rightfully pointed out is less effective than H&E).

In "conclusion", I would correct the statement "Ex vivo FCM provides digital high-resolution imaging of leukoplakia that is comparable to histopathology", as FCM is (slightly at least) less effective than H&E.

Author Response

Dear Reviewer,

thank you for your time and effort and for giving us the opportunity to submit a revised manuscript. We have incorporated the suggestions you made. Those changes are highlighted within the manuscript.

Please find below a point-by-point response to your comments and concerns.

  1. The paper - which I have already reviewed in your previous submission - has been significantly improved and is of a certain interest for a journal such as diagnostic. Study design and methods are adequate. It should be accepted in my opinion but I would improve/enlarge the discussion part in which author hypothesize future application for this technique (which, as they rightfully pointed out is less effective than H&E).

Author response:

Thank you. As suggested, we enlarged the discussion with a focus on cost saving potential.

  1. In "conclusion", I would correct the statement "Ex vivo FCM provides digital high-resolution imaging of leukoplakia that is comparable to histopathology", as FCM is (slightly at least) less effective than H&E.

Author response:

Thank you. We corrected the statement as suggested

Kind regards,

Veronika Shavlokhova

Reviewer 2 Report

Congratulations to the authors : allow to the reviewer these two remarks which should be taken into account before editing the final manuscript

  • please give the exact reference of the last WHO definition of the oral leukoplakia; your reference n° 3 is not exactly what you means...)
  • Line 52 : about the "low" cost of this new technology, the authors should at least mention in the discussion and conclusion some words about the cost of such a kind of laser and microscope. Parallell to this, they should compare the cost of a classical oral biopsy + histopathology exam. Although these values may be very different in different locations, a comparaison of the cost pf the two technics in the same hospital or in the sale ambulant structure should at least give a good idea to the reader

Author Response

Dear Reviewer,

thank you for your time and effort and for giving us the opportunity to submit a revised manuscript. We have incorporated the suggestions you made. Those changes are highlighted within the manuscript.

Please find below a point-by-point response to your comments and concerns.

Kind regards,

Veronika Shavlokhova

This manuscript is a resubmission of an earlier submission. The following is a list of the peer review reports and author responses from that submission.